# Genetic Diversity and Trends of Ancestral and New Inbreeding in Deutsch Drahthaar Assessed by Pedigree Data

**DOI:** 10.3390/ani12070929

**Published:** 2022-04-05

**Authors:** Paula Wiebke Michels, Ottmar Distl

**Affiliations:** Institute of Animal Breeding and Genetics, University of Veterinary Medicine Hannover (Foundation), 30559 Hannover, Germany; paula.wiebke.michels@tiho-hannover.de

**Keywords:** Deutsch Drahthaar dogs, genetic diversity, inbreeding, ancestral inbreeding, probability of gene origin, pedigree analysis

## Abstract

**Simple Summary:**

Deutsch Drahthaar (DD) is the most popular hunting dog in Germany, fulfilling all aspects of hunting including searching for trails. This breed was newly created at the beginning of the 20th century from a large number existing versatile hunting dog breeds. The aim of the breed was, and still is, to achieve the best performance in all aspects of hunting. We analyzed pedigrees of DD using demographic measures to quantify genetic diversity such as probabilities of gene origin and degrees of ancestral and individual inbreeding. A large number of genetically diverse founder dogs should open up the opportunity of creating a breed with a high genetic diversity and a low increase of inbreeding per generation. On the other hand, intense use of top sires and dams from a limited number of breeding lines may accelerate breeding progress in hunting abilities but reduce genetic diversity. Monitoring genetic diversity should help to maintain a high diversity of breeding populations. Our analysis of pedigree data from 101,887 DD dogs revealed inbreeding measures (coefficient of inbreeding F = 0.042, individual rate of inbreeding ΔF_i_ = 0.00551) and effective population size (*Ne* = 92) in the mean range compared to a wide range of other dog breeds. Ancestral inbreeding had a strong increasing trend, whereas trends in individual inbreeding and rate of individual inbreeding were slightly negative.

**Abstract:**

Loss of genetic diversity and high inbreeding rates confer an increased risk of congenital anomalies and diseases and thus impacting dog breeding. In this study, we analyzed recent and ancestral inbreeding as well as other measures of genetic variability in the Deutsch Drahthaar (DD) dog population. Analyses included pedigree data from 101,887 animals and a reference population with 65,927 dogs born between 2000 and 2020. The mean equivalent complete generations was 8.6 with 69% known ancestors in generation 8. The mean realized effective population size was 92 with an increasing trend from 83 to 108 over birth years. The numbers of founders, effective founders and effective ancestors, as well as founder genomes, were 814, 66, 38 and 16.15, respectively. Thirteen ancestors explained 50% of the genetic diversity. The mean coefficient of inbreeding and individual rate of inbreeding (ΔF_i_) were 0.042 and 0.00551, respectively, with a slightly decreasing trend in ΔF_i_. Exposure of ancestors to identical-by-descent alleles explored through ancestral coefficients of inbreeding showed a strong increasing trend. Comparisons between new and ancestral inbreeding coefficients according to Kalinowski et al. showed an average relative contribution of 62% of new inbreeding to individual inbreeding. Comparisons among average coancestry within the parental population and average inbreeding in the reference population were not indicative of genetic substructures. In conclusion, the creation of the DD dog breed about 120 years ago resulted in a popular breed with considerable genetic diversity without substructuring into lines or subpopulations. The trend of new inbreeding was declining, while ancestral inbreeding through ancestors who were autozygous at least once in previous generations was increasing.

## 1. Introduction

Hunting is of great importance for humans as well as for the environment. Although hunting is no longer the basis of the food supply, it is necessary for the promotion of wildlife species and thus for nature and animal conservation and it serves to represent the status quo [1,2]. According to German legislation, well-trained hunting dogs must be available for hunting (§ 4 NJagdG). One of the most popular versatile hunting dogs is the Deutsch Drahthaar dog (DD) with approximately 3000 puppies born per year in the German dog breeding association for DD, Verein für Deutsch Drahthaar (VDD) under the German parent organization for dog husbandry (Verband für das Deutsche Hundewesen e.V.) [3]. The foundation of the DD began as early as 1902 in Germany with the aim of adopting the best dogs from existing versatile hunting dog breeds [4]. The idea of creating a new breed from intercrosses of other versatile hunting dog breeds with similar genetic origins, rather than improving an existing pure dog breed, was taken up by Edvard Karel Korthals at the end of the 19th century with the founding of the Griffon club in 1883 [5]. Based on the ideas of Korthals and Hegewald, the DD movement began in 1902 to breed only with dogs that had proven their performance, leading after 25 years to a popular hunting dog breed with a growing number of breeders [6,7]. The Pudelpointer, Griffon Korthals, Deutsch Stichelhaar (German Broken Coated Pointing Dog) and Deutsch Kurzhaar (German Shorthaired Pointer, SHP) were reported as the founding breeds [6]. The principles of breeding DD aimed at all aspects of a versatile hunting dog for field, forest and water work as well as retrieving lost game and searching for trails of wounded game. The DD was officially recognized in Germany in 1927, when the breed had developed into an independent and popular hunting dog [4]. In order to avoid show breeding, a breed standard was not established until 1969 [4]. Once the breed standard was established, the Fédération Cynologique Internationale (FCI) officially recognized the DD as a pointing dog breed under the standard number 98 [8]. In North America, a distinction must be made between dogs bred under the VDD, which are called DD dogs, and German Wire-haired Pointers (GWP). GWP are descended from imported DD dogs registered with the American Kennel Club, but bred for a long time without restrictions and regulations and can therefore be distinguished from DD dogs [9].

Intensive selection for specific traits such as hunting performance and morphological phenotypes can lead to bottlenecks and thus, increased inbreeding rates. Consequently, the probability of homozygosity for recessive genes can lead to a higher prevalence of hereditary diseases and reduced fitness traits [10,11]. To control for these effects, the evaluation of pedigree data to calculate measures of genetic diversity is an important issue. 

Pedigree-based inbreeding coefficients (F) vary widely among different dog breeds [12,13,14] (Appendix A). Czech Spotted Dogs (F = 0.36), Nova Scotia Duck Tolling Retrievers (F = 0.26), American dog guides (F = 0.15/0.26) and Polish Lowland Sheepdogs (F = 0.18) had particularly high inbreeding coefficients [15,16,17,18]. Other highly inbred dog breeds included Polish Hunting Dogs and Greyhounds, Lancashire Heeler, Ibizan hounds, Bichon frise, Lundehund and Hungarian Border Collies [13,16,19,20,21,22]. To our knowledge, no pedigree analysis of genetic diversity and inbreeding measures was conducted for DD dogs. Few data are available for the DD foundation breeds SHP and Griffon. Leroy et al. [12] included the SHP (F = 0.035) and the Griffon Korthals (F = 0.056) in a genetic diversity study of 60 French dog populations. Italian SHPs had an F of 0.023 [23]. Because classical inbreeding coefficients do not take into account the degree of inbreeding in ancestors, ancestral inbreeding coefficients were developed some time ago [24,25,26]. However, ancestral inbreeding coefficients have rarely been estimated in dog breeding [22]. Another approach that provides insights into genetic variability through the use of gene origin probabilities, which take into consideration founder and ancestor contributions and allow detection of bottlenecks and genetic drift in populations [27,28], has been used more frequently (Appendix A).

The objective of this study was to evaluate the genetic diversity of the DD population born from 2000 to 2020. Therefore, we analyzed the probabilities of gene origin, classical and ancestral inbreeding coefficients and effective population size based on pedigree data containing 101,887 DDs.

## 2. Materials and Methods

Pedigree data were provided by the VDD. Data included all available pedigree records of the VDD population. The pedigree file employed for analysis contained 101,887 animals with 7881 dams and 3782 sires. For the analysis of demographic measures, only animals with both known parents were included. To reduce the risk of underestimation of the parameters, only birth years with at least 6.5 GE were included in the reference population. Therefore, dogs with both parents known and born between 2000 and 2020 were defined as the reference population. On average 3139 animals were born per birth year. The reference population comprised 65,927 animals descending from 4501 dams and 2124 sires. The founders in the pedigrees available were born in the 1960s.

Pedigree analyses were carried out using the PEDIG software [29]. Data editing and calculation of individual rates of inbreeding and effective population size were done using SAS, version 9.4 (Statistical Analysis System, Cary, NC, USA, 2021).

The number of equivalent complete generations (GE) was calculated to allow an assessment of the pedigree completeness. GE was defined as the sum of the proportion of known ancestors over all generations [30].

Animals with unknown ancestors, to which all other individuals of the population could be traced back, formed the founders (*f*) and were expected to be unrelated with an inbreeding coefficient of zero. As a measure of genetic diversity in the breed and the number of effective founders (*f_e_*) was estimated as: (1)$$f_{e}=\frac{1}{\sum_{k=1}^{f}q_{\begin{array}{c} k \end{array}}^{2}}$$
with *f* = number of founders, *q_k_* = probability of gene origin of the individual ancestor (*k*) [27]. This value indicates the number of equally contributing founders.

The effective number of ancestors (*f_a_*) was calculated to assess how balanced the use of reproductive animals was. We calculated *f_a_* as the marginal genetic contribution (*q*) of an individual ancestor (*q_i_*) and thus the contribution of an ancestor (*a*) that could not be explained by any other ancestor before [27]:(2)$$f_{a}=\frac{1}{\sum_{j=1}^{a}q_{j}^{2}}$$

The effective number of founder genomes (*f_g_*) was chosen to describe the balanced contribution of founders and random losses of founder alleles in descendants. It was estimated in the same way as the effective number of founders, but the individual contribution of founders was replaced by the contribution of founder genes [28]:(3)$$f_{g}=\frac{1}{\sum_{k=1}^{f}\frac{q_{k}^{2}}{r_{\begin{array}{c} k \end{array}}}}$$
with *f* = number of founders, *q_k_* = proportion of genes of the founder *k,* which can be found in the reference population and *r_k_* = expected proportion of founder alleles that have been kept within the descendant population.

Ratios *f_a_*/*f_e_* and *f_g_*/*f_e_* were used to indicate the loss of genetic diversity during the development since the founder generation. *F_a_*/*f_e_*-values < 1 refer to bottlenecks in the population. The impact of drift on the population is shown by the *f_g_*/*f_e_*-ratio.

The amount of genetic diversity (*GD*), which accounts for the loss of genetic diversity resulting from genetic drift and unequal contribution of founders, was calculated using the following formula [31]:(4)$$GD=1-\frac{1}{2f_{g}}$$

Correspondingly, the amount of genetic diversity (*GD**) accounting for loss of genetic diversity due to an unequal contribution of founders was estimated as follows [31]:(5)$$GD^{*}=1-\frac{1}{2f_{e}}$$

The loss of genetic diversity as a consequence of genetic drift can be expressed as the difference between *GD** and *GD*.

The inbreeding coefficient according to Meuwissen and Luo [32] was estimated for the whole reference population (F) and separately for inbred dogs (F_inbred_).

Individual inbreeding using the genedrop method (F_gd_), measures of ancestral inbreeding according to Ballou [24] (F_a_Bal_), ancestral (F_a_Kal_) and new (F_New_) inbreeding coefficient according to Kalinowski et al. [25] and an ancestral history coefficient defined by Baumung et al. [26] (AHC) were estimated using the GRAIN package, version 2.2 [26,33]. 

The ancestral inbreeding coefficient F_a_Bal_ is defined as the cumulative proportion of an individual’s genome which was previously exposed to ancestral inbreeding [24]. F_a_Bal_ considers inbreeding from each ancestor and increases with each inbred ancestor in the pedigree and is independent of the individual inbreeding coefficient [24]. Thus, individuals with an individual inbreeding coefficient of zero can have a F_a_Bal_ > 0. A gradual increase of F_a_Bal_ compared to F over the years implies an increase of inbred ancestors in the pedigrees, but not necessarily an increase of individual inbreeding. 

According to Kalinowski et al. [25], inbreeding coefficients can be divided into ancestral and new inbreeding. F_a_Kal_ refers to the currently homozygous alleles that were already homozygous in at least one ancestor of the individual. F_New_ represents those alleles that are currently homozygous for the first time. Because ancestors in common have to occur on both sides of the pedigree in inbred individuals, F_a_Kal_ is zero if the individual inbreeding coefficient F is zero [25]. 

The ancestral history coefficient AHC according to Baumung et al. measures how often a randomly taken allele was in IBD status during pedigree segregation [26].

Trends of the different inbreeding coefficients were quantified by an analysis by birth years. 

Pearson correlation coefficients among the different coefficients of inbreeding were calculated using SAS, version 9.4 (Statistical Analysis System, Cary, NC, USA, 2021). In addition, we calculated Pearson correlation coefficients for individual and ancestral inbreeding coefficients between offspring and parents as well as between both parents using SAS. 

In order to assess the genetic substructure of the reference population we compared the average coancestry within the parental population (Φ) with the average coefficient of inbreeding according to Meuwissen and Luo [32] of the reference population for all birth years and by birth year cohorts. The degree of deviation of random mating from Hardy–Weinberg proportions was estimated as [34]:(6)$$\propto =1-\frac{1-F}{1-\Phi }$$

The individual rate of inbreeding (ΔF_i_) was calculated dependent on the GE according to Gutiérrez et al. [35] to adjust for the pedigree depth in the calculations:(7)$$\Delta F_{i}=1-\sqrt[{\;GE_{i}-1}]{1-F_{i}}.$$

Based on the mean of ΔF_i_ ($\bar{\Delta F_{i}\;}$) the realized effective population size (*N_e_*) of the DD reference population was estimated [36]:(8)$$N_{e}=\frac{1}{2\bar{\Delta F_{i}\;}}.$$

The effective population size (*N_e_*) represents the number of reproducing animals in an idealized population that would produce the same genetic diversity as the population under study. The expected effective population size under random mating (*N_ec_*) was calculated accordingly using Φ instead of F and mean GE of both parents [36]. The expected effective population size from the increase of coancestry among parents per GE corresponds to the number of reproducing animals in an idealized population under random mating.

The unbalanced use of reproducers can be estimated through the effective number of sires (*NeffS*) and dams (*NeffD*):(9)$$NeffS=\frac{1}{\sum_{i}s_{i}{}^{2}}\;\mathrm{and}\;NeffD=\frac{1}{\sum_{i}d_{i}{}^{2}},$$
where *s_i_* (*d_i_*) is the relative frequency of use of the sire or dam *i* among all sires (dams) of the reference population [37].

## 3. Results

Mean GE was 8.62 and increased steadily from 6.5 to 10.3 in the birth years 2000 to 2020. The mean proportion of known ancestors of the reference population in generations 4, 6, 8 and 10 was 95%, 89%, 69% and 38%, respectively. The numbers of founders (*f*) and the effective number of founders (*f_e_*) and founder genomes (*f_g_*) were 814, 66, and 16.2 (Table 1). *N_e_* and ΔF_i_ were 91.6 and 0.00551. Ratios of *f_e_*/*f*, *f_a_*/*f_e_*, and *f_g_*/*f_e_* were lower than 1. While *f_g_* decreased from 2000 to 2020, *f*, *f_e_* and *f_a_* decreased until 2011–2015 and then increased again (Appendix A). 

The amount of genetic diversity lost in the reference population since founder generation due to bottlenecks and genetic drift (1 − *GD*) was 0.031 and losses due to genetic drift were 0.0233 (*GD** − *GD*). Losses through unequal contributions of founders (1 − *GD**) reached 0.008. The relative effects due to genetic drift on genetic diversity were 75.9% and thus were much larger than the influences due to unequal use of founders with 24.1%. 

The cumulated marginal contributions of the 10 and 15 most contributing ancestors were 0.43 and 0.54, respectively (Table 2).

The average inbreeding coefficient was slightly higher for inbred than for all DD dogs (Table 3). The average ancestral inbreeding coefficients AHC and F_a_Bal_ were the highest inbreeding coefficients. The ancestral inbreeding coefficient F_a_Kal_ was lower than the new inbreeding F_New_.

The correlation between the ancestral inbreeding coefficients F_a_Bal_ and AHC was 1 and F_a_Kal_ had high correlations with F and F_gd_ (Table 4). Inbreeding coefficients F and F_gd_ were identical. F_New_ had very low correlations with AHC and F_a_Bal_, but high correlations with F and F_gd_ as well as moderate correlations with F_a_Kal_. Correlations of inbreeding coefficients between both parents as well as between parents and offspring were low (Appendix A). 

Comparison among average coancestry within parents and average individual coefficients of inbreeding in the reference population gave an α-value of 0.0153 and α-values of 0.0133 to 0.0135 for the three birth year cohorts from 2000–2015 and 0.0062 for the birth year cohort 2016–2020 (Table 5). The largest differences between *N_e_* and *N_ec_* were found in the 2000–2005 birth year cohort and the smallest in the 2016–2020 birth year cohort.

The proportion of inbred animals in the DD reference population fluctuated between 94.1 and 99.5% in the birth years 2000 to 2016 and decreased in the following birth year to 83.3%, followed by an increase to 91.8% in 2020 (Figure 1). The mean coefficient of inbreeding per birth year increased steadily from 0.032 in the birth year 2000 to 0.050 in 2015 and dropped down to 0.040 in 2017, but increased again in the following birth years to 0.047 in 2019.

The ancestral coefficients of inbreeding AHC and F_a_Bal_ showed a strong increasing trend over time (Figure 2, Appendix A). A slight increase was seen for F_New_ and F_a_Kal_ up to the birth year 2017, and then a small decline was found for both coefficients.

The mean individual rate of inbreeding per generation was highest in the birth years 2000 and 2001, and between 2002 and 2016, ΔF_i_ varied between 0.0054 and 0.0059 (Figure 3, Appendix A). In 2017, ΔF_i_ decreased to 0.0044 and increased to 0.0050 in 2019. *N_e_* was inversely related to the individual rate of inbreeding per generation, with the lowest estimate of 79.3 in 2001 and the highest estimate of 114.2 in 2017.

## 4. Discussion

In this study, the development of the genetic diversity of the DD population including birth years between 2000 and 2020 was analyzed using demographic measures, individual rates of inbreeding, and ancestral, new and conventional coefficients of inbreeding. Pedigree depth was with an estimate of 8.6 GE in the higher range compared to other studies (Appendix A). Completeness and quality of pedigree data are of major importance for the estimation of pedigree-based inbreeding measures and data should be more than four GE [12,38]. Incomplete pedigrees may lead to underestimated inbreeding measures [27,38,39].

*N_e_* was calculated using the individual rate of inbreeding and thus was independent of the number of known generations in the pedigree [35]. The *N_e_* of DD dogs was in the upper range when compared with other dog breeds (Appendix A). Higher values had a Swedish population of Rottweilers, and French populations of English Setters and English Cocker Spaniels with *N_e_* of 134, 128 and 123 [12,40]. Similar to DD, German Dalmatians, French Yorkshire Terriers and Australian Cavalier King Charles Spaniels had an *N_e_* of about 90 [12,13,41]. For Nova Scotia Duck Tolling Retrievers, Hungarian Border Collies and Polish Lowland Sheepdogs, quite small estimates of about 20 were reported [16,18,22]. The *N_e_* of Griffon Korthals (*N_e_* = 49) was lower and the *N_e_* of SHPs (*N_e_* = 92) was similar to the DD population studied [12]. The *N_e_* is an important parameter for the evaluation of the genetic diversity and the endangerment status of populations. *N_e_* in the range of 50 to 100 is considered critical for the endangerment of populations. Bijma [42] suggested a ΔF_i_ of 0.5 to 1.0% per generation, which corresponds to an *N_e_* of 50 to 100. The DD population exceeded this threshold. Even if the mean *N_e_* for the complete reference population was below 100, *N_e_* for the birth years from 2017 to 2020 was above this critical value. The European Association for Animal Production set a threshold of 5% inbreeding in 50 years for a non-endangered population [43]. With an estimated increase in inbreeding of 6.06% in 50 years, this threshold will be missed. However, considering the increasing trend of the *N_e_* in the recent birth years, it can be assumed that this threshold will be reached. 

The reason for the great genetic diversity of this relatively new hunting dog breed may be seen in the broad use of a large number of genetically diverse founders, the increasing popularity resulting in a large population size and the breeding management aiming at a restricted use of sires and dams. The average and median number of progeny per sire were 32 and 14, respectively, with 8 to 38 progeny and 4 to 120 in the 25% to 75% and 5% to 95% quantiles, respectively. There were only 26 popular sires with more than 200 progeny. Effective sires and effective dams in the reference population were 702 and 2763, respectively, as well as per birth year at 132 ± 10 and 359 ± 32, respectively (Appendix A). Ratios between effective and observed numbers of sires and dams per birth year of 0.59 and 0.86 indicated balanced use of sires and dams. Furthermore, we found no indications of line breeding or a meaningful increase of inbreeding in offspring of parents already inbred as correlations of individual and ancestral measures of inbreeding among parents were very low in the DD reference population. Similarly, F_New_ of the animal had a very low correlation to F_a_Kal_ and other measures of ancestral inbreeding of the dam and sire indicative that new inbreeding in the offspring was not associated with ancestral inbreeding of parents. Analysis of probabilities of gene origin indicated only a moderate influence on loss of genetic diversity through unequal use of founder animals even if a large proportion of genetic diversity was lost through genetic drift in consequence of a severe decrease of an effective number of founder genomes compared to an effective number of founders. Parameters derived from the probabilities of gene origin help to describe the development of the genetic variability in a population. Dividing the reference population into four birth year classes showed a decreasing trend for *f_g_* and *f_g_*/*f_e_* due to an increasing genetic drift. Founders *f_e_*, *f_a_* and *f_a_*/*f_e_* slightly increased in 2016–2020, and inbreeding measures decreased due to a more balanced representation of founders in the reference population of these birth years. The founders of the 2016–2020 reference population did not lead to an increase of *f_g_*, as these dogs were not introgressed from other dog breeds into DD, but came from lines that were not used as frequently or regularly in breeding during 2005–2015. Thus, the original gene pool of DD was not increased. A slight increase of *f_e_* and *f_a_* in 2016–2020 indicated a more balanced representation of founders than in the 5–10 birth years before. 

The number of founders was reasonably high in the DD population. Cavalier King Charles and English Cocker Spaniels achieved similar levels with 836 and 837 founders, respectively [13]. Nova Scotia Duck Tolling Retrievers were based on only 19 founders [16], while 2286 founders were reported for a Belgian population of German Shepherd dogs (GSD) [44]. The effective number of founders in the DD population was strongly declining compared to the number of founders, resulting in a *f_e_*/*f*-ratio of 0.08. A total of 32 Australian breeds had an averaged *f_e_*/*f*-ratio of 0.18 with populations with very large registries (>80.000) or a larger GE (>8) comparable to the DD dogs having smaller mean ratios of 0.14 and 0.17 [13]. An Australian population of GSD and Australian Working Kelpies had the smallest ratios with 0.05 and 0.06, respectively [13,45]. With a *f_e_*/*f*-ratio of 0.52, Nova Scotia Duck Tolling Retrievers showed large values [16]. The small *f_e_*/*f*-ratio indicates a strong unequal contribution of founders and the presence of genetic drift in the early development of the population. The *f_a_*/*f_e_*-ratio of 0.58 in DD suggests the presence of bottlenecks since the founder generation due to an unbalanced use of breeding animals. Despite some populations such as the Bavarian Mountain dogs (*f_a_*/*f_e_* = 0.78) [46] or the Lancashire Heeler (*f_a_*/*f_e_* = 0.89) [16] having only moderate recent bottlenecks, many dog populations had more severe bottlenecks (Appendix A). One Hungarian Border Collie population had a *f_a_*/*f_e_*-ratio of 0.17 [22]. The impact of the loss of genetic diversity due to random genetic drift, independent of founder contributions, was high in DD, but similar to many other dog populations (Appendix A). Australian breeds with a GE > 8, and thus comparable to the DD dogs under study, had lower *f_g_*/*f_e_*-ratios (0.20) than the DD population studied here. An Australian population of 44,396 Labrador Retrievers had a very similar *f_g_*/*f_e_*-ratio [13]. Lower losses of genetic diversity due to random genetic drift were especially found in Hungarian Border Collies [22] and Finish Spitz [47], while Hanoverian Hounds [46] and several Swedish dog populations such as grey Norwegian Elkhounds [40] among others had higher losses of genetic diversity due to random genetic drift compared to DD dogs. 

Further indicators for breeding management are coefficients of inbreeding. In the DD, F was at the same level as in many other dog breeds (Appendix A). SHPs had a slightly lower F [12,23] and Griffon Korthals a higher F compared to the DD population under study [12]. The DD population had an average ΔF_i_ estimate of 0.00551, which is in the lower range compared with many other dog breeds (Appendix A). All mean values for ΔF_i_ across breeds exceeded the estimate for the DD population. The large population size of the DD population with a large number of genetically diverse founders may have contributed for the low estimates of F and ΔF_i_. Correlations of individual, new and ancestral inbreeding measures between both parents were low, indicating that matings among highly inbred partners and partners with inbred ancestors in their pedigrees were uncommon. In addition, a comparison of average coancestry within parents with average coefficients of inbreeding in the reference population revealed that clustering of the population was largely avoided and thus, the increase of inbreeding was kept at a low rate. The restriction of matings for sires with a maximum of 6 litters per year seemed to prevent genetic substructures and line breeding. In the 2016–2020 birth year cohort, α decreased to 0.00624 reflecting a slightly increasing trend of average coancestry among parents but a stronger opposite trend in the individual coefficient of inbreeding. Therefore, the decrease in α may be explained by fewer matings among close relatives and thus, by the avoidance of new inbreeding, as shown by lower estimates for F_New_ and the individual rate of inbreeding per generation (Appendix A). Accordingly, the differences between *N_e_* and *N_ec_* were smallest for the 2016–2020 birth year cohort. 

To assess the risk of negative effects of high inbreeding coefficients, it is important to distinguish between ancestral and recent inbreeding. Recent inbreeding is assumed to be more harmful than ancient inbreeding, because the frequency of deleterious alleles is expected to decrease over time through selection [48]. To our knowledge, Hungarian Border Collies were the only other dog population in which ancestral inbreeding has been analyzed to date. In the latter population, ancestral and new inbreeding developed in the same way over the years and new inbreeding was attributable to approximately 35% of individual inbreeding in the last year of birth [22]. In the DD population, a slightly decreasing trend of F over the birth years appeared to be independent of ancestral inbreeding. F_a_Bal_ and AHC showed a strong increasing trend, indicating an increasing number of ancestors in the pedigree who were autozygous at least once in previous generations. The correlation of 1 between F_a_Bal_ and AHC indicated that these coefficients have similar assumptions, despite the different estimation methods. Therefore, estimation of only one of the two coefficients may be sufficient. The inbreeding coefficients by Kalinowski et al. [25] showed that ancestral inbreeding increased over time, but new inbreeding increased up to 2011 and then decreased below the level in 2000. Regarding the ancestral inbreeding coefficients, much of the inbreeding in the DD population was due to past breeding practices. However, on average, 62% of the individual inbreeding in the reference population was caused by recent and thus more harmful inbreeding. From this perspective, new inbreeding in particular should be further reduced in order to minimize the risk associated with inbreeding. Therefore, in addition to F, inbreeding coefficients according to Kalinowski et al. [25] should be used in planning future matings. For a more detailed monitoring of the population, ancestral inbreeding should be evaluated in addition to individual inbreeding in order to avoid a further increase of new inbreeding and to reduce the impact of ancestral inbreeding in future generations. In particular, breeding management can prevent the number of matings between individuals from leading to an increase in new inbreeding and reduce the number of matings among individuals whose ancestors were autozygous at least once in previous generations. In addition, limiting the number of matings of sires and dams is useful in maintaining a large proportion of the effective number of founders and founder genomes.

## 5. Conclusions

The results of this study indicate a loss of genetic diversity mainly due to genetic drift, as shown by the loss of effective founder genomes in comparison to the effective number of founders. The effective number of founder genomes continued to decrease and the number of ancestors that were autozygous in previous generations increased in the pedigrees. The ancestral inbreeding coefficients indicate a loss of genetic diversity in the past due to the use of breeding animals with common ancestors. Despite unequal employment of breeding animals and genetic drift in the last 60 years, a large genetic diversity has been maintained and the individual rate of inbreeding has been kept at a low level in the last 20 years. 

## Figures and Tables

**Figure 1 animals-12-00929-f001:**
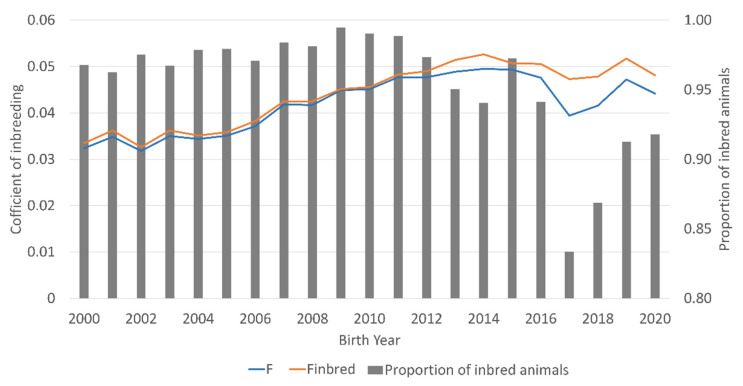
Inbreeding coefficients for all (F) and only inbred (F_inbred_) Deutsch Drahthaar dogs as well as the proportion of inbred animals for the birth years 2000 to 2020.

**Figure 2 animals-12-00929-f002:**
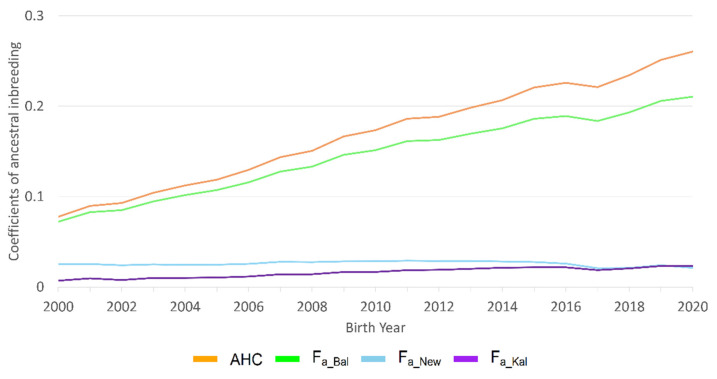
Ancestral inbreeding according to Ballou (F_a_Bal_), ancestral (F_a_Kal_) and new (F_New_) inbreeding according to Kalinowski et al. and the ancestral history coefficient according to Baumung et al. (AHC) of Deutsch Drahthaar dogs for the birth years 2000 to 2020.

**Figure 3 animals-12-00929-f003:**
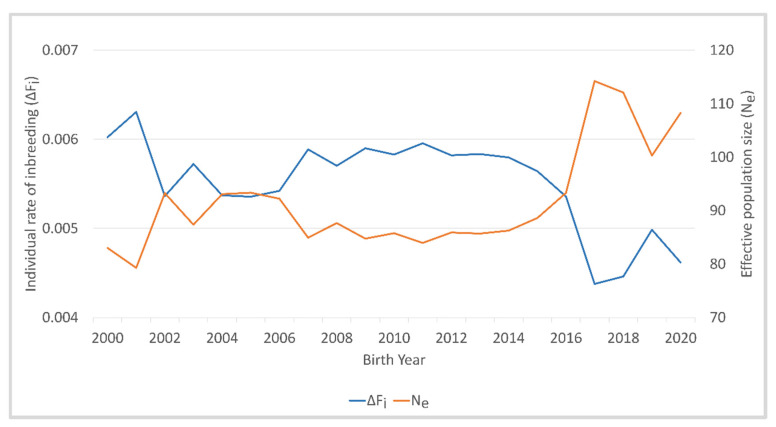
Individual rate of inbreeding (ΔF_i_) and effective population size (*N_e_*) of Deutsch Drahthaar dogs for the birth years 2000 to 2020.

**Table 1 animals-12-00929-t001:** Pedigree analysis for measures of genetic diversity for Deutsch Drahthaar dogs born from 2000 to 2020.

Parameter	Reference Population from 2000 to 2020
Reference population	65,927
Inbred animals in reference population	62,984
Mean equivalent generations (GE)	8.62
Mean generation interval in years	4.42
Number of founders (*f*)	814
Effective number of founders (*f_e_*)	65.5
Effective number of ancestors (*f_a_*)	37.8
Effective number of founder genomes (*f_g_*)	16.2
*f_e_*/*f*	0.08
*f_a_*/*f_e_*	0.58
*f_g_*/*f_e_*	0.25
Ancestors explaining 30% of the gene pool	5
Ancestors explaining 40% of the gene pool	8
Ancestors explaining 50% of the gene pool	13
Ancestors explaining 60% of the gene pool	19
Ancestors explaining 70% of the gene pool	29
Ancestors explaining 80% of the gene pool	49
Ancestors explaining 90% of the gene pool	110
Ancestors explaining 95% of the gene pool	177
Effective population size (*N_e_*)	91.6
ΔF_i_	0.00551

**Table 2 animals-12-00929-t002:** Ancestors with the greatest marginal contributions onto the Deutsch Drahthaar reference population.

ID	Birth Year	Sex	Marginal Contribution	Cumulated Marginal Contributions	Number of Progeny
112374	1980	male	0.0846	0.0846	117
144135	1989	male	0.0577	0.1422	218
70425	1969	female	0.0573	0.1995	16
87288	1973	female	0.0407	0.2402	28
145543	1989	female	0.0362	0.2765	34
106677	1978	female	0.0358	0.3123	8
134384	1986	male	0.0350	0.3473	155
129824	1984	female	0.0286	0.3759	46
90647	1974	female	0.0272	0.4031	7
79470	1971	male	0.0266	0.4296	10
110712	1979	male	0.0243	0.4539	52
90556	1974	male	0.0237	0.4776	11
109594	1979	male	0.0213	0.4989	7
105465	1978	male	0.0206	0.5195	21
94096	1975	male	0.0172	0.5367	8

**Table 3 animals-12-00929-t003:** Mean inbreeding coefficients according to Meuwissen and Luo for all animals (F) and inbred animals (F_inbred_), individual inbreeding using the genedrop method (F_gd_), ancestral inbreeding according to Ballou (F_a_Bal_) and Kalinowski et al. (F_a_Kal_), new inbreeding according to Kalinowski (F_New_) and ancestral history coefficient according to Baumung et al. (AHC), for Deutsch Drahthaar dogs born from 2000 to 2020.

Method	Inbreeding Coefficients
F	0.042
F_inbred_	0.044
F_gd_	0.042
F_a_Bal_	0.145
F_a_Kal_	0.016
F_New_	0.026
AHC	0.168

**Table 4 animals-12-00929-t004:** Pearson correlation coefficients between inbreeding coefficient according to Meuwissen and Luo (F), individual inbreeding using the genedrop method (F_gd_), ancestral inbreeding according to Ballou (F_a_Bal_) and Kalinowski et al. (F_a_Kal_), new inbreeding according to Kalinowski (F_New_) and ancestral history coefficient according to Baumung et al. (AHC) for Deutsch Drahthaar dogs born from 2000 to 2020. *p*-values of all correlation coefficients were <0.0001.

Inbreeding Coefficients	F	F_gd_	F_a_Bal_	F_a_Kal_	F_New_	AHC
F		1.00	0.41	0.87	0.95	0.40
F_gd_			0.41	0.87	0.95	0.40
F_a_Bal_				0.68	0.18	1.00
F_a_Kal_					0.67	0.68
F_New_						0.16

**Table 5 animals-12-00929-t005:** Average coancestry within the parental population (Φ), average individual inbreeding coefficient according to Meuwissen and Luo (F), degree of deviation (α) of random mating from Hardy–Weinberg proportions, realized effective population size (*N_e_*) and expected effective population size under random mating (*N_ec_*) for the reference population and by birth year cohorts.

Population	Average Coancestry within Parents (Φ)	F	Deviation from Random Mating (α)	*N_e_*	*N_ec_*
Reference	0.027	0.042	0.01528	92	139
2000–2005	0.021	0.034	0.01328	88	143
2006–2010	0.029	0.042	0.01339	87	127
2011–2015	0.036	0.049	0.01345	86	115
2016–2020	0.038	0.044	0.00624	105	116

## Data Availability

Restrictions apply to the availability of these data. Data were obtained from the Verein Deutsch Drahthaar e.V. and are on reasonable request available from the authors with the permission of the Verein Deutsch Drahthaar e.V.

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
