# Peer review of "Genetic Diversity and Trends of Ancestral and New Inbreeding in Deutsch Drahthaar Assessed by Pedigree Data"

_animals, 2022, doi:10.3390/ani12070929_

Round 1
Reviewer 1 Report
Concerning the manuscript I am glad to realize that authors accepted most of my comments and suggestions. As a result the scientific level of the manuscript has been significantly improved. I found the methods appropriate, the results were clearly presented and discussed. I also agree with the conclusions. I suggest the publishment of the manuscript in its present form.
Author Response
Reviewer 1
We thank reviewer 1 for the work and time to improve our manuscript.
Open Review
(x) I would not like to sign my review report
( ) I would like to sign my review report
English language and style
( ) Extensive editing of English language and style required
( ) Moderate English changes required
( ) English language and style are fine/minor spell check required
(x) I don't feel qualified to judge about the English language and style
|
Yes |
Can be improved |
Must be improved |
Not applicable |
|
|
Does the introduction provide sufficient background and include all relevant references? |
(x) |
( ) |
( ) |
( ) |
|
Is the research design appropriate? |
(x) |
( ) |
( ) |
( ) |
|
Are the methods adequately described? |
(x) |
( ) |
( ) |
( ) |
|
Are the results clearly presented? |
(x) |
( ) |
( ) |
( ) |
|
Are the conclusions supported by the results? |
(x) |
( ) |
( ) |
( ) |
Comments and Suggestions for Authors
Concerning the manuscript I am glad to realize that authors accepted most of my comments and suggestions. As a result the scientific level of the manuscript has been significantly improved. I found the methods appropriate, the results were clearly presented and discussed. I also agree with the conclusions. I suggest the publishment of the manuscript in its present form.
Response: no changes requested.
Reviewer 2 Report
In this new version of the article “Genetic Diversity of Deutsch Drahthaar Assessed by Pedigree Data”, I consider that most of my comments have been considered, and I have only minor suggestions that authors can be trusted to consider.
Table 5 indicate a substantial decrease of the deviation from Random mating in 2016-2020 in comparison to the 3 former cohorts. Yet, this point is not discussed. The data presented here show a deviation first in the range of what is found with other dog breeds (see Fis results from Leroy et al. 2013, even if indicator is slightly different), and then much less deviation for the last cohort. Could this trend be related to lower substructure within the breed? To less mating between close relatives? Please note that this could be linked to the decrease in deltaFi/increase in Ne observed in the period.
L271-272. “the estimation of pedigree-based inbreeding measures and data should be more than four GE” please either add references or justify why four GE.
L292 Maybe “relatively newly created”? The creation of DD is not that new compared to multiple breeds that have been created over the last few decades…
Author Response
Reviewer 2
We thank reviewer2 for time and labor to improve our manuscript.
Open Review
(x) I would not like to sign my review report
( ) I would like to sign my review report
English language and style
( ) Extensive editing of English language and style required
( ) Moderate English changes required
( ) English language and style are fine/minor spell check required
(x) I don't feel qualified to judge about the English language and style
|
Yes |
Can be improved |
Must be improved |
Not applicable |
|
|
Does the introduction provide sufficient background and include all relevant references? |
(x) |
( ) |
( ) |
( ) |
|
Is the research design appropriate? |
(x) |
( ) |
( ) |
( ) |
|
Are the methods adequately described? |
(x) |
( ) |
( ) |
( ) |
|
Are the results clearly presented? |
(x) |
( ) |
( ) |
( ) |
|
Are the conclusions supported by the results? |
( ) |
(x) |
( ) |
( ) |
Comments and Suggestions for Authors
In this new version of the article “Genetic Diversity of Deutsch Drahthaar Assessed by Pedigree Data”, I consider that most of my comments have been considered, and I have only minor suggestions that authors can be trusted to consider.
Table 5 indicate a substantial decrease of the deviation from Random mating in 2016-2020 in comparison to the 3 former cohorts. Yet, this point is not discussed. The data presented here show a deviation first in the range of what is found with other dog breeds (see Fis results from Leroy et al. 2013, even if indicator is slightly different), and then much less deviation for the last cohort. Could this trend be related to lower substructure within the breed? To less mating between close relatives? Please note that this could be linked to the decrease in deltaFi/increase in Ne observed in the period.
Amended:
Line 188-192: The expected effective population size under random mating (Nec) was calculated accordingly using Φ instead of F and mean GE of both parents [36]. The expected effective population size from the increase of coancestry among parents per GE corresponds to the number of reproducing animals in an idealized population under random mating.
Line 241-242: The largest differences between Ne and Nec were found in the 2000-2005 birth year cohort and the smallest in the 2016-2020 birth year cohort.
Line 243-247: Table 5. Average coancestry within the parental population (Φ), average individual inbreeding coefficient according to Meuwissen and Luo (F), degree of deviation (α) of random mating from Hardy-Weinberg proportions, realized effective population size (Ne) and expected effective population size under random mating (Nec) for the reference population and by birth year cohorts.
In Table 5 two columns were added to make visible differences in effective population size under the given mating system and a random mating system.
Line 369-375: In the 2016-2020 birth year cohort, α decreased to 0.00624 reflecting a slightly increasing trend of average coancestry among parents but a stronger opposite trend in the individual coefficient of inbreeding. Therefore, the decrease in α may be explained by fewer matings among close relatives and thus, by the avoidance of new inbreeding, as shown by lower estimates for FNew and the individual rate of inbreeding per generation (Table S2). Accordingly, the differences between Ne and Nec were smallest for the 2016-2020 birth year cohort.
L271-272. “the estimation of pedigree-based inbreeding measures and data should be more than four GE” please either add references or justify why four GE.
Amended:
Line 279: we inserted two references for justifying 4 GE, namely [12,39].
L292 Maybe “relatively newly created”? The creation of DD is not that new compared to multiple breeds that have been created over the last few decades…
Amended: We agree with this comment.
Line 300: The reason for the great genetic diversity of this relatively newly created hunting dog
This manuscript is a resubmission of an earlier submission. The following is a list of the peer review reports and author responses from that submission.
Round 1
Reviewer 1 Report
The topic of the manuscript is relevant and falls to the scope of the journal. Detailed knowledge of the population's structure under selection is vital for the breeder. Genealogy records (if they are long and complete) provide valuable sources of information for this type of analysis. The authors analyzed the Deutsch Drahthaar dog population which so far has not been analyzed by means of pedigree analysis. The applied methods were adequate but concerning methods I have some suggestions which will improve the scientific quality. I agree with most conclusions, but also I have some suggestions. My specific suggestions are as follows.
2. Materials and Methods
The exact time of the whole period was not given (only the reference period). Authors has to provide in which year the most remote dog was born and when the youngest animal was born.
3. Results
The results given in table 1 has to be given also for the whole examination period. On the other hand it does not make sense to give several numbers (e.g. for CGE it ranges from 6.5 to 10.3) for the reference period. Regarding tables 1-3 do not mention exact numbers which were already given in the tables. Bessides for Table 3 define what Fgd was. Concerning figures 1-2-3, again it does not make sense to give pattern for reference population. It is generally characterized by one number. In you think otherwise it only means that your reference population is too wide (i.e. cca 4-5 generation). Re-defining reference population into several separate reference populations can be considered (e.g. 2000-2005, 2006-2010, 2011-2015, 2016-2020).
I request that Figures 1-3 should be plotted based on the whole examination period. Table 4. is very important and should be discussed much longer. I suggest that authors should mention that due to the correlation 1.0 between AHC and FBallou applying one or the other is sufficient. I also would emphasize that FNEW is very important and its use is preferable against F. Altogether my impressions were positive and I kindly ask authors to accepts my suggestions in order to further improve the scientific quality of the paper.
Author Response
Dear Mrs. Sandra Spatariu
thank you for your time and labour to handle our manuscript. We thank the reviewers for their valuable and useful comments.
We revised the manuscript accordingly.
Best regards
Ottmar
Dear Dr. Distl,
Thank you for submitting the following manuscript to Animals:
Manuscript ID: animals-1134260
Type of manuscript: Article
Title: Genetic Diversity of Deutsch Drahthaar Assessed by Pedigree Data
Authors: Paula Wiebke Michels, Ottmar Distl *
Received: 18 February 2021
E-mails: paula.wiebke.michels@tiho-hannover.de, ottmar.distl@tiho-hannover.de
Submitted to section: Animal Genetics and Genomics,
https://www.mdpi.com/journal/animals/sections/Animal_Genetics_and_Genomics
It has been reviewed by experts in the field and we request that you make
major revisions before it is processed further. Please find your manuscript
and the review reports at the following link:
https://susy.mdpi.com/user/manuscripts/resubmit/d03519edab19bd5a7390259a1127ac96
Your co-authors can also view this link if they have an account in our
submission system using the e-mail address in this message.
Please revise the manuscript according to the reviewers' comments and upload
the revised file within 10 days. Use the version of your manuscript found at
the above link for your revisions, as the editorial office may have made
formatting changes to your original submission. Any revisions should be
clearly highlighted, for example using the "Track Changes" function in
Microsoft Word, so that changes are easily visible to the editors and
reviewers. Please provide a cover letter to explain point-by-point the
details of the revisions in the manuscript and your responses to the
reviewers' comments. Please include in your rebuttal if you found it
impossible to address certain comments. The revised version will be inspected
by the editors and reviewers. Please detail the revisions that have been
made, citing the line number and exact change, so that the editor can check
the changes expeditiously. Simple statements like ‘done’ or ‘revised as
requested’ will not be accepted unless the change is simply a typographical
error.
Please carefully read the guidelines outlined in the 'Instructions for
Authors' on the journal website
https://www.mdpi.com/journal/animals/instructions and ensure that your
manuscript resubmission adheres to these guidelines. In particular, please
ensure that abbreviations have been defined in parentheses the first time
they appear in the abstract, main text, and in figure or table captions;
citations within the text are in the correct format; references at the end of
the text are in the correct format; figures and/or tables are placed at
appropriate positions within the text and are of suitable quality; tables are
prepared in MS Word table format, not as images; and permission has been
obtained and there are no copyright issues.
If the reviewers have suggested that your manuscript should undergo extensive
English editing, please have the English in the manuscript thoroughly checked
and edited for language and form. Alternatively, MDPI provides an English
editing service checking grammar, spelling, punctuation and some improvement
of style where necessary for an additional charge (extensive re-writing is
not included), see details at https://www.mdpi.com/authors/english.
Do not hesitate to contact us if you have any questions regarding the
revision of your manuscript or if you need more time. We look forward to
hearing from you soon.
Kind regards,
Sandra Spatariu
Section Managing Editor, MDPI
E-mail: spatariu@mdpi.com
Reviewer 1
Open Review
(x) I would not like to sign my review report
( ) I would like to sign my review report
English language and style
( ) Extensive editing of English language and style required
( ) Moderate English changes required
(x) English language and style are fine/minor spell check required
( ) I don't feel qualified to judge about the English language and style
|
|
Yes |
Can be improved |
Must be improved |
Not applicable |
|
Does the introduction provide sufficient background and include all relevant references? |
(x) |
( ) |
( ) |
( ) |
|
Is the research design appropriate? |
(x) |
( ) |
( ) |
( ) |
|
Are the methods adequately described? |
( ) |
(x) |
( ) |
( ) |
|
Are the results clearly presented? |
( ) |
(x) |
( ) |
( ) |
|
Are the conclusions supported by the results? |
( ) |
(x) |
( ) |
( ) |
Comments and Suggestions for Authors
The topic of the manuscript is relevant and falls to the scope of the journal. Detailed knowledge of the population's structure under selection is vital for the breeder. Genealogy records (if they are long and complete) provide valuable sources of information for this type of analysis. The authors analyzed the Deutsch Drahthaar dog population which so far has not been analyzed by means of pedigree analysis. The applied methods were adequate but concerning methods I have some suggestions which will improve the scientific quality. I agree with most conclusions, but also I have some suggestions. My specific suggestions are as follows.
- Materials and Methods
The exact time of the whole period was not given (only the reference period). Authors has to provide in which year the most remote dog was born and when the youngest animal was born.
Amended:
Line 112: The founders in the pedigrees available were born in the 1960s.
- Results
The results given in table 1 has to be given also for the whole examination period. On the other hand it does not make sense to give several numbers (e.g. for CGE it ranges from 6.5 to 10.3) for the reference period.
Amended, comment: Values were estimated for the reference population and not for the whole population to enable a comparable quality of pedigree data for the compared birth years. A sentence describing the choice of the reference population has been added to M&M:
LL 107-109: In order to reduce the risk of underestimation of the parameters, only birth years with at least 6.5 GE were included in the reference population. Therefore, dogs with both parents known and born from 2000 to 2020 were defined as reference population.
In addition, estimates for further birth years and birth year cohorts are given in Table S2.
Regarding tables 1-3 do not mention exact numbers which were already given in the tables.
Amended, comment:
Lines 182-222:
Exact numbers already mentioned in tables were deleted.
Besides for Table 3 define what Fgd was.
Amended, new lines:
LL 152: “Individual inbreeding using the genedrop method (Fgd), measures of ancestral inbreeding according to Ballou [23] (Fa_Bal), ancestral (Fa_Kal) and new (FNew) inbreeding coefficient according to Kalinowski, et al. [24] and an ancestral history coefficient defined by Baumung, et al. [25] (AHC) were estimated using the GRAIN package, version 2.2. [25,32].”
Concerning figures 1-2-3, again it does not make sense to give pattern for reference population. It is generally characterized by one number. In you think otherwise it only means that your reference population is too wide (i.e. cca 4-5 generation). Re-defining reference population into several separate reference populations can be considered (e.g. 2000-2005, 2006-2010, 2011-2015, 2016-2020). I request that Figures 1-3 should be plotted based on the whole examination period.
Comment: Values were estimated for the reference population and not for the whole population to enable a comparable quality of pedigree data for the compared birth years. A sentence describing the choice of the reference population has been added to M&M:
LL 107-109 “To reduce the risk of underestimation of the parameters, only birth years with at least 6.5 GE were included in the reference population. Therefore, dogs with both parents known and born from 2000 to 2020 were defined as reference population.”
Line 256-257: Completeness and quality of pedigree data is of major importance for the estimation of pedigree-based inbreeding measures and data should exceed 4 GE.
In addition, estimates for further birth years and birth year cohorts are given in Table S2.
Table 4. is very important and should be discussed much longer. I suggest that authors should mention that due to the correlation 1.0 between AHC and FBallou applying one or the other is sufficient. I also would emphasize that FNEW is very important and its use is preferable against F.
Amended, new lines:
LL 258-359:
Therefore, estimation of only one of both coefficients may be sufficient.
LL 365-367: Thus, inbreeding coefficients according to Kalinowski, et al. [25] should be used besides F in planning future matings.
Altogether my impressions were positive and I kindly ask authors to accepts my suggestions in order to further improve the scientific quality of the paper.
Reviewer 2 Report
In the article “Genetic Diversity of Deutsch Drahthaar Assessed by Pedigree Data” authors analyse the genetic variability of a dog breed based on pedigree information. The scope is restricted to one single breed, which has not be investigated until now. Approaches used are mostly adequate although improvable. There are however subsequent improvement required in terms of background information, concept presented and some methodological choice. As a consequence, authors conclusion is most certainly wrong.
First, I have been surprised on the information provided on the history of breeds and the populations presented as founder breeds.
For instance, Griffon is a general term for bearded dog breeds which often have nothing to do within each other. In the case of Drahtaar it most probably refers to Griffon Korthals (which by the way has been investigated in some studies using pedigree data), but Griffon bleu de Gascogne and Griffond Belge and Bruxellois are certainly not part of the founder breeds of the Drahtaar. It is unclear to me why authors decided to focus their comparative analysis on founding breeds, but in any case, those three breeds are not part of it.
Another example, given the fact that for instance Griffon Korthals creation is based on a cross between several breeds around 1860, presenting the intercross that led to the creation of Drahtaar as “revolutionary” is a complete non-sense.
In terms of methodology, the fact to choose an effective population size metrics based on inbreeding increase rate maybe an issue since the Gutierrez indicator is both sensible to (i) population structure and (ii) variation in pedigree knowledge (see Leroy et al. 2020). In the paper, authors never investigate if there is some population structure in the breed studied, for instance by comparing average inbreeding and coancestry. Besides, the variation observed in 2017 in inbreeding clearly relates to variation in pedigree knowledge (and probably entry of new founders), which is, first, not discussed, and, secondly, may likely have affected the trend in Ne, and therefore the full conclusion of the paper.
More emphasis on this issue is clearly needed.
Various comments:
L8-21 Simple summary need to be rewritten: most of it content is quite vague and do not refer to the study by itself and there is no pertinent results presented. Quantifying genetic diversity is redundant with probability of gene origin and inbreeding.
L23 Similarly authors seems to suggest that recent and ancestral inbreeding are not measures of genetic variability.
L46-47 This is quite vague… available to who?
L53 this is untrue (see above)
L68-69 This basically depend of the intensity of the selection.
L72-73 pedigree analysis go beyond the calculation of demographic measures.
L80-83 as stated above, if authors want to focus on a comparison with founding breeds, please focus on Korthals and not on breed which have nothing to do.
L262-263 as stated by authors, average generation equivalent, is around 8, which means that, supposing a generation interval around 4-5 years, the genealogical founders of the file trace back on average to 40-50 years ago, i.e. are certainly not the historical founders of the breed. Besides I would not qualified as newly created, a breed created more than 100 years ago. Eventually authors could refer to regular entry of unrelated animals, however, this suppose that genealogical founders are really unrelated, which is unlikely to be the case. In absence of information on the origin of the founder which led to the decrease of inbreeding in 2017. I would certainly not write such a conclusion.
L295 actually they aren’t, or to be more precise coefficients of inbreeding need to be interpreted with caution, considering the different kind of bias (pedigree knowledge, population structure) affecting them.
L333 authors cannot prove it, as most of their founders are certainly not the ones used for the creation of the breed.
Author Response
Dear Mrs. Sandra Spatariu
thank you for your time and labour to handle our manuscript. We thank the reviewers for their valuable and useful comments.
We revised the manuscript accordingly.
Best regards
Ottmar
Reviewer 2
Open Review
(x) I would not like to sign my review report
( ) I would like to sign my review report
English language and style
( ) Extensive editing of English language and style required
( ) Moderate English changes required
(x) English language and style are fine/minor spell check required
( ) I don't feel qualified to judge about the English language and style
|
|
Yes |
Can be improved |
Must be improved |
Not applicable |
|
Does the introduction provide sufficient background and include all relevant references? |
(x) |
( ) |
( ) |
( ) |
|
Is the research design appropriate? |
( ) |
( ) |
(x) |
( ) |
|
Are the methods adequately described? |
(x) |
( ) |
( ) |
( ) |
|
Are the results clearly presented? |
( ) |
( ) |
(x) |
( ) |
|
Are the conclusions supported by the results? |
( ) |
( ) |
(x) |
( ) |
Comments and Suggestions for Authors
In the article “Genetic Diversity of Deutsch Drahthaar Assessed by Pedigree Data” authors analyse the genetic variability of a dog breed based on pedigree information. The scope is restricted to one single breed, which has not be investigated until now. Approaches used are mostly adequate although improvable. There are however subsequent improvement required in terms of background information, concept presented and some methodological choice. As a consequence, authors conclusion is most certainly wrong.
First, I have been surprised on the information provided on the history of breeds and the populations presented as founder breeds.
For instance, Griffon is a general term for bearded dog breeds which often have nothing to do within each other. In the case of Drahtaar it most probably refers to Griffon Korthals (which by the way has been investigated in some studies using pedigree data), but Griffon bleu de Gascogne and Griffond Belge and Bruxellois are certainly not part of the founder breeds of the Drahtaar. It is unclear to me why authors decided to focus their comparative analysis on founding breeds, but in any case, those three breeds are not part of it.
Amended, comment: Values for Griffon bleu de Gascogne, Belge and Bruxellois were deleted in both, introduction and discussion. Values for Griffon Korthals were added.
New lines:
LL 88-91: “Leroy, et al. [11] included the SHP (F=0.035) and the Griffon Korthals (F=0.056) in a study of genetic diversity of 60 French dog populations.” LL “The realized Ne depended on GE of Griffon Korthals (Ne=65) was lower or in similar range depended on the breed definition and the Ne of SHPs (Ne=92) was similar to the DD population under study [12].” LL “SHPs had a slightly lower F [11,22] and Griffon Korthals a higher F compared to the DD population under study [11].”
Another example, given the fact that for instance Griffon Korthals creation is based on a cross between several breeds around 1860, presenting the intercross that led to the creation of Drahthaar as “revolutionary” is a complete non-sense.
Amended, new lines:
LL 53-63:
The idea to create a new breed from intercrosses of other versatile hunting dog breeds with a similar genetic origin instead of improving an existing pure dog breed was introduced by Edvard Karel Korthals in the late 19th century with foundation of the Griffon club in 1883 [5]. Based on the ideas of Korthals and Hegewald, the DD movement to breed only with dogs under proven performance started in 1902 and resulted in a popular hunting dog breed with an increasing number of breeders after 25 years [6,7].
In terms of methodology, the fact to choose an effective population size metrics based on inbreeding increase rate maybe an issue since the Gutierrez indicator is both sensible to (i) population structure and (ii) variation in pedigree knowledge (see Leroy et al. 2020). In the paper, authors never investigate if there is some population structure in the breed studied, for instance by comparing average inbreeding and coancestry. Besides, the variation observed in 2017 in inbreeding clearly relates to variation in pedigree knowledge (and probably entry of new founders), which is, first, not discussed, and, secondly, may likely have affected the trend in Ne, and therefore the full conclusion of the paper.
More emphasis on this issue is clearly needed.
Amended, new lines:
LL 283-287:
Furthermore, we found no indications of substructures or line breeding in the DD reference population as correlations of individual and ancestral measures of inbreeding among parents were very low. Similarly, FNew of the animal had a very low correlation to Fa_Kal and other measures of ancestral inbreeding of the dam and sire indicative that new inbreeding in the offspring was not related to ancestral inbreeding of parents.
LL 292-301:
Subdivision of the reference population into four birth year classes showed a decreasing trend for fg and the ratio fg/fe due to an increasing genetic drift. Founders, fe, fa and fa/fe slightly increased in 2016-2020 and inbreeding measures decreased due to a larger number of founders of DD bred in these birth years. The founders of the 2016-2020 reference population did not lead to an increase of fg because these dogs were not introgressed from other dog breeds into DD but were from lines not so often or not regularly employed in the years 2005-2015 in breeding. Thus, the original gene pool of DD was not enlarged. A slight increase of fe and fa in 2016-2020 indicated a more balanced use of founders than in the 5-10 birth years before.
LL 340-346:
Correlations of inbreeding measures between both parents were low, showing that matings among highly inbred partners were uncommon. In addition, clustering of the population was largely avoided and thus, increase of inbreeding was kept on a low rate.
LL370-375:
Individual rate of inbreeding showed a decreasing and Ne an increasing trend for the last birth years due to a slight increase in founders of the DD breed. However, effective number of founder genomes were still decreasing.
Comment: Corresponding values are given in Table S3.
Various comments:
L8-21 Simple summary need to be rewritten: most of it content is quite vague and do not refer to the study by itself and there is no pertinent results presented. Quantifying genetic diversity is redundant with probability of gene origin and inbreeding.
LL18-20:
Our analysis of pedigree data from 101,887 DD dogs showed inbreeding measures (coefficient of inbreeding F=0.042, individual rate of inbreeding DFi=0.00551) and effective population size (Ne=92) in the mean range compared to a wide range of other dog breeds.
L23 Similarly authors seems to suggest that recent and ancestral inbreeding are not measures of genetic variability.
Amended, new lines:
LL 25: “In this study, we analyzed recent and ancestral inbreeding as well as other measures of genetic variability of the Deutsch Drahthaar (DD) dog population.”
L46-47 This is quite vague… available to who?
Amended, new lines: LL 47-48: “According to German legislation, well trained hunting dogs must be available for hunting (§ 4 NJagdG).”
L53 this is untrue (see above)
Amended, new lines: LL 53-56:
The idea to create a new breed from intercrosses of other versatile hunting dog breeds with a similar genetic origin instead of improving an existing pure dog breed was introduced by Edvard Karel Korthals in the late 19th century with foundation of the Griffon club in 1883 [5].
L68-69 This basically depend of the intensity of the selection.
Amended, new lines: LL 75-80: “Intensive selection for special characteristics like hunting performance and phenotypes can may result in bottlenecks and thus increasing inbreeding rates.”
L72-73 pedigree analysis go beyond the calculation of demographic measures.
Amended, new lines: LL 79-80: “To control these effects, evaluation of pedigree data for calculation of demographic measures of genetic diversity is an important issue.”
L80-83 as stated above, if authors want to focus on a comparison with founding breeds, please focus on Korthals and not on breed which have nothing to do.
Amended, new lines: LL “Leroy, et al. [11] included the SHP (F=0.035) and the Griffon Korthals (F=0.056) in a study of genetic diversity of 60 French dog populations.“
L262-263 as stated by authors, average generation equivalent, is around 8, which means that, supposing a generation interval around 4-5 years, the genealogical founders of the file trace back on average to 40-50 years ago, i.e. are certainly not the historical founders of the breed. Besides I would not qualified as newly created, a breed created more than 100 years ago. Eventually authors could refer to regular entry of unrelated animals, however, this suppose that genealogical founders are really unrelated, which is unlikely to be the case. In absence of information on the origin of the founder which led to the decrease of inbreeding in 2017. I would certainly not write such a conclusion.
Comment:
In DD, there are not regularly entering unrelated dogs. There is strong regulation for breeding. It was a new philosophy to create the DD breed. In the first 20 years after founding the club, there were strong controversial discussions and all other breeders rejected these dogs as bastards or cess-pool or phantom-dogs. To resist all these hostilities and at the end to be successful was a great pioneering performance. The work of Korthals also flowed into the DD in Germany; otherwise the heritage of this breed may have been lost in Germany, in France Korthals Griffons experienced their revival in 1950s and 1960s.
See above: we changed the introduction referring to breed foundation.
LL 373-374 this sentence was deleted: “A high number of founders contributed to the creation of the DD population about 120 years ago.“
L295 actually they aren’t, or to be more precise coefficients of inbreeding need to be interpreted with caution, considering the different kind of bias (pedigree knowledge, population structure) affecting them.
Amended, new lines: LL “Good Further indicators for the breeding management are coefficients of inbreeding.”
L333 authors cannot prove it, as most of their founders are certainly not the ones used for the creation of the breed.
LL374-377:
Loss of genetic diversity was mainly due to genetic drift resulting from an unequal use of founders in the reference population. Individual rate of inbreeding showed a decreasing and Ne an increasing trend for the last birth years due to a slight increase in founders of the reference population.
Round 2
Reviewer 2 Report
In this new version of the article “Genetic Diversity of Deutsch Drahthaar Assessed by Pedigree Data” authors improved the manuscript, however there is still a number of issues that require changes. I do not think that analysis used here can be used to support the absence of structure, while a simple comparison of inbreeding vs coancestries would have allow to do so. Also, the data set used here, although quite complete, is far from representing the history of the breed since its creation. Therefore, authors cannot make any link between the way the breed was created and its current genetic variability.
Various comments :
L53-58 Again, the passage is misleading. The source does not indicate that crossbreeding was an idea first brought by E. Korthals. Actually while in the XIXth century, purebreeding as a practice became slowly the norm, crossbreeding between populations of different or similar origins was nothing like a new concept (see e.g. the “Traité de Vènerie” from d’Yauville, 1788).
L169-171 please be more precise. Measures of genetic diversity between individuals would correspond to metrics such as genetic distance or coancestries. Authors should indicate there are referring to correlations between measures of genetic diversity.
L270-274 Absence of correlation between inbreeding and ancestral inbreeding does not ncessarely means that there is no structure within the population. It just means that breeding practices leading to recent inbreeding (for instance mating between close relatives) are disconnected from old sources of inbreeding.
L322-323 The interpretation of the sentence is wrong. The fact that there is little correlations in inbreeding between parents does not preclude inbreeding to their offspring, as inbreeding of offspring is not the sum of inbreeding of the parents. To assess the clustering of the population, authors should have compare the average coancestry between parents and the average inbreeding of offspring, which has not been done.
L359-361 Also misleading. Given the pedigree knowledge, the huge majority of founders considered here do not trace back to the creation of the breed. This sentence can therefore not be supported by the results of the analysis.
Author Response
Manuscript ID: animals-1134260
Dear Sandra Spatariu
Thank you for handling our manuscript and your time and work.
We thank the reviewers for their input to improve our manuscript. We revised the manuscript according to their recommendations.
Kind regards
Ottmar Distl
Dear Dr. Distl,
Thank you for submitting the following manuscript to Animals:
Manuscript ID: animals-1134260
Type of manuscript: Article
Title: Genetic Diversity of Deutsch Drahthaar Assessed by Pedigree Data
Authors: Paula Wiebke Michels, Ottmar Distl *
Received: 18 February 2021
E-mails: paula.wiebke.michels@tiho-hannover.de, ottmar.distl@tiho-hannover.de Submitted to section: Animal Genetics and Genomics, https://www.mdpi.com/journal/animals/sections/Animal_Genetics_and_Genomics
It has been reviewed by experts in the field and we request that you make major revisions before it is processed further. Please find your manuscript and the review reports at the following link:
https://susy.mdpi.com/user/manuscripts/resubmit/d03519edab19bd5a7390259a1127ac96
Your co-authors can also view this link if they have an account in our submission system using the e-mail address in this message.
Please revise the manuscript according to the reviewers' comments and upload the revised file within 5 days. Use the version of your manuscript found at the above link for your revisions, as the editorial office may have made formatting changes to your original submission. Any revisions should be clearly highlighted, for example using the "Track Changes" function in Microsoft Word, so that changes are easily visible to the editors and reviewers. Please provide a cover letter to explain point-by-point the details of the revisions in the manuscript and your responses to the reviewers' comments. Please include in your rebuttal if you found it impossible to address certain comments. The revised version will be inspected by the editors and reviewers. Please detail the revisions that have been made, citing the line number and exact change, so that the editor can check the changes expeditiously. Simple statements like ‘done’ or ‘revised as requested’ will not be accepted unless the change is simply a typographical error.
Please carefully read the guidelines outlined in the 'Instructions for Authors' on the journal website https://www.mdpi.com/journal/animals/instructions and ensure that your manuscript resubmission adheres to these guidelines. In particular, please ensure that abbreviations have been defined in parentheses the first time they appear in the abstract, main text, and in figure or table captions; citations within the text are in the correct format; references at the end of the text are in the correct format; figures and/or tables are placed at appropriate positions within the text and are of suitable quality; tables are prepared in MS Word table format, not as images; and permission has been obtained and there are no copyright issues.
If the reviewers have suggested that your manuscript should undergo extensive English editing, please have the English in the manuscript thoroughly checked and edited for language and form. Alternatively, MDPI provides an English editing service checking grammar, spelling, punctuation and some improvement of style where necessary for an additional charge (extensive re-writing is not included), see details at https://www.mdpi.com/authors/english.
Do not hesitate to contact us if you have any questions regarding the revision of your manuscript or if you need more time. We look forward to hearing from you soon.
Kind regards,
Sandra Spatariu
Section Managing Editor, MDPI
E-mail: spatariu@mdpi.com
--
MDPI Branch Office, Cluj-Napoca
Office: Avram Iancu 454, Cluj-Napoca, Romania
Disclaimer: MDPI recognizes the importance of data privacy and protection. We treat personal data in line with the General Data Protection Regulation
(GDPR) and with what the community expects of us. The information contained in this message is confidential and intended solely for the use of the individual or entity to whom they are addressed. If you have received this message in error, please notify me and delete this message from your system.
You may not copy this message in its entirety or in part, or disclose its contents to anyone.
Formularbeginn
Open Review
(x) I would not like to sign my review report
( ) I would like to sign my review report
English language and style
( ) Extensive editing of English language and style required
( ) Moderate English changes required
(x) English language and style are fine/minor spell check required
( ) I don't feel qualified to judge about the English language and style
|
Yes |
Can be improved |
Must be improved |
Not applicable |
|
|
Does the introduction provide sufficient background and include all relevant references? |
(x) |
( ) |
( ) |
( ) |
|
Is the research design appropriate? |
( ) |
(x) |
( ) |
( ) |
|
Are the methods adequately described? |
(x) |
( ) |
( ) |
( ) |
|
Are the results clearly presented? |
(x) |
( ) |
( ) |
( ) |
|
Are the conclusions supported by the results? |
( ) |
( ) |
(x) |
( ) |
Comments and Suggestions for Authors
In this new version of the article “Genetic Diversity of Deutsch Drahthaar Assessed by Pedigree Data” authors improved the manuscript, however there is still a number of issues that require changes. I do not think that analysis used here can be used to support the absence of structure, while a simple comparison of inbreeding vs coancestries would have allow to do so. Also, the data set used here, although quite complete, is far from representing the history of the breed since its creation. Therefore, authors cannot make any link between the way the breed was created and its current genetic variability.
Amended:
We added assessment of population structure using coancestries among parents and comparisons with measures of inbreeding.
We removed the sentence linking breed history to the data of the reference population analysed here.
Various comments :
L53-58 Again, the passage is misleading. The source does not indicate that crossbreeding was an idea first brought by E. Korthals. Actually while in the XIXth century, purebreeding as a practice became slowly the norm, crossbreeding between populations of different or similar origins was nothing like a new concept (see e.g. the “Traité de Vènerie” from d’Yauville, 1788).
Amended:
Line 53-58:
The idea to create a new breed from intercrosses of other versatile hunting dog breeds with a similar genetic origin instead of improving an existing pure dog breed was employed by Edvard Karel Korthals in the late 19th century with foundation of the Griffon club in 1883 [5].
L169-171 please be more precise. Measures of genetic diversity between individuals would correspond to metrics such as genetic distance or coancestries. Authors should indicate there are referring to correlations between measures of genetic diversity.
Amended:
Line 169-174:
Pearson correlation coefficients among the different coefficients of inbreeding were calculated using SAS, version 9.4 (Statistical Analysis System, Cary, NC, USA, 2021). In addition, we calculated Pearson correlation coefficients for individual and ancestral inbreeding coefficients between offspring and parents as well as among both parents using SAS. To assess clustering of the reference population we estimated average coancestries between parents and correlated average parental coancestries with coefficients of inbreeding in offspring.
L270-274 Absence of correlation between inbreeding and ancestral inbreeding does not ncessarely means that there is no structure within the population. It just means that breeding practices leading to recent inbreeding (for instance mating between close relatives) are disconnected from old sources of inbreeding.
Amended:
Line 276-278
Furthermore, we found no indications of line breeding or a meaningful increase of inbreeding in offspring of parents already inbred as correlations of individual and ancestral measures of inbreeding among parents were very low in the DD reference population.
L322-323 The interpretation of the sentence is wrong. The fact that there is little correlations in inbreeding between parents does not preclude inbreeding to their offspring, as inbreeding of offspring is not the sum of inbreeding of the parents. To assess the clustering of the population, authors should have compare the average coancestry between parents and the average inbreeding of offspring, which has not been done.
Amended:
Line 173-175
To assess clustering of the reference population we estimated average coancestries between parents and correlated average parental coancestries with coefficients of inbreeding in offspring.
Line 214-215
Similarly, correlations among average coancestries among parents and coefficients of inbreeding in offspring were close to zero (Table S3).
Line 328-333
Correlations of individual, new and ancestral inbreeding measures between both parents were low, showing that matings among highly inbred partners and partners with inbred ancestors in their pedigrees were uncommon. In addition, comparison of average coancestries between parents with coefficients of inbreeding in offspring revealed that clustering of the population was largely avoided and thus, increase of inbreeding was kept on a low rate.
Line 378-380
Table S3. Correlations among measures of inbreeding of dogs included in the reference population with inbreeding measures of their sires and dams as well as parental average coancestry (a) and in addition, correlations of inbreeding measures among parents (b).
L359-361 Also misleading. Given the pedigree knowledge, the huge majority of founders considered here do not trace back to the creation of the breed. This sentence can therefore not be supported by the results of the analysis.
Amended:
Line 357-368
This sentence was removed in Conclusions.
Creation of a new breed using a large number of founders from different existing breeds with large genetic distances proved to be a successful concept for the DD population.
Submission Date
18 February 2021
Date of this review
17 Mar 2021 09:22:46
Formularende
© 1996-2021 MDPI (Basel, Switzerland) unless otherwise stated
Round 3
Reviewer 2 Report
|
Author Response
Manuscript ID: animals-1134260
Dear Sandra Spatariu
Thank you for handling our manuscript and your time and work.
We thank the reviewer for input to improve our manuscript and the valuable comment on the relation among coancestry averaged across all parents and inbreeding in offspring. We revised the manuscript according to these recommendations.
Kind regards
Ottmar Distl
In this third round of review, authors indicate they computed coancestries as suggested, I am however puzzled by the results presented in Table S3.
It is indeed very unclear what has been computed. Coancestry between two parents corresponds to the inbreeding of their offspring. Therefore, the correlation computed on each individuals, and the corresponding parents pairs is expected to be 1. On the other hand, authors refers to “Averaged coancestries among parents”, averaged over what? If it is among all the parents pairs of the reference population, then how authors did compute a correlation?
Sorry, for this misunderstanding from us. Correlation among coancestry of parents with their corresponding offspring was 1.
Assessing existence of a genetic substructure is actually quite simple. Authors just need to compare the average inbreeding in the reference population, and the average coancestry within the parental population (i.e. not considering the actual pairs, but all possible combination). As the number of coancestries can be quite large (n*(n-1)/2 with n being the number of parents), authors can eventually make a random sampling of pairs of parents. Lower average coancestry within the parental population compared to average inbreeding within the offspring population indicate deviation from Hardy-Weinberg equilibrium related to genetic structure within the population, i.e. either presence of subpopulations (wahlund effect), or existence of practice such as line or close-breeding (i.e. intentional mating between close relatives).
Amended:
Line 178-181
To assess clustering of the reference population we compared the average coancestry among the parental population (F) with the average coefficient of inbreeding in offspring. The degree of deviation of random matings from Hardy-Weinberg proportions was estimated as [34]:
|
(6) |
Line 215-216
Mean coancestry in parental population was 0.027 resulting in an a of 0.0154.
Line 352-366
In addition, comparison of average coancestry between all parents of the reference population with the average inbreeding coefficient in offspring revealed a discrepancy from random mating indicative for clustering of the population or matings among close relatives. Deviations from Hardy-Weinberg proportions were much higher in French SHPs (a=0.0333), whereas in French Griffon Korthals with an a=0.0136 smaller. Across 60 French dog breeds, a was at 0.0153 and thus, very similar to the estimate in DD studied here [12]. An example for a high deviation from random mating related to genetic structure within population or line breeding were Rough Collies with an a of 0.0541 [14].
Another remarks. ΔFi is not a measure of inbreeding coefficient but of inbreeding rate. It should therefore not appear as such in table 3.
Comment: we moved ΔFi intoTable 1.